# CONFORMAL RISK-AVERSE DECISION MAKING WITH ACTION CONDITIONAL GUARANTEES

## ABSTRACT

Reliable decision making pipelines powered by machine learning models require uncertainty quantification (UQ) methods that come with explicit safety guarantees. Conformal prediction provides such UQ by wrapping ML predictions into prediction sets, and recent work by Kiyani et al. (2025) established that these sets can be translated into optimal risk-averse decision policies—yet only inheriting marginal safety guarantees. We generalize and strengthen their results by (i) introducing action-conditional conformal prediction, which yields safety guarantees conditioned explicitly on each action taken by the decision maker, (ii) showing that action-conditional prediction sets serve as a proxy for the feasible decision space for risk-averse decision makers aiming to optimize action-conditional value-at-risk, and (iii) proposing a principled finite-sample algorithm based on pinball-loss minimization, connecting the framework of Gibbs et al. (2025) to action-conditional guarantees. Experiments on two real-world datasets confirm that our approach significantly improves action-conditional performance over several conformal baselines.

## 1 INTRODUCTION

In many high-stakes scenarios—such as clinical medicine, finance, or autonomous systems—machine learning models are increasingly deployed to assist in decision making. Consider a healthcare application, where a predictive model is suggesting a serious condition based on a patient data. Underestimating the model's predictive uncertainty could lead to recommending a high-risk treatment that results in catastrophic outcomes. Overestimating the uncertainty, on the other hand, could lead to overly conservative decisions that forgo potentially life-saving interventions. This illustrates a fundamental trade-off in practice: balancing safety, by guarding against low-utility outcomes, with utility, by leveraging predictive information to maximize gains.

Conformal prediction (CP) has emerged as a central framework for uncertainty quantification in supervised learning, offering distribution-free, model-agnostic guarantees with efficient post-hoc implementations. In the standard setup, let $(X, Y) \sim \mathcal{D}$ be a random pair with feature $X \in \mathcal{X}$ and label $Y \in \mathcal{Y}$. CP constructs prediction sets $C(X) \subseteq \mathcal{Y}$ that satisfy a marginal coverage guarantee: $\mathbb{P}(Y \in C(X)) \geq 1 - \alpha$. Despite its desirable statistical properties, it remains less clear how such sets should guide principled downstream decision making. Recent progress by Kiyani et al. (2025) has advanced this connection by studying *risk-averse decision makers*—agents who, upon observing $X$ but not $Y$, must choose an action $a \in \mathcal{A}$ to maximize a utility function $u : \mathcal{A} \times \mathcal{Y} \to \mathbb{R}$, while ensuring that low-utility outcomes occur with probability at most $\alpha$. Specifically, given an action policy $a : \mathcal{X} \to \mathcal{A}$ and a utility certificate $\nu : \mathcal{X} \to \mathbb{R}$, they consider the problem of *maximizing the expected certificate* $\mathbb{E}[\nu(X)]$ subject to the *safety constraint*,

$$\mathbb{P}(u(a(X), Y) \geq \nu(X)) \geq 1 - \alpha. \tag{1}$$

This is a *marginal safety guarantee*, meaning the agent is safe on average over the distribution of covariates—for example, on average across patients in a medical setting. Kiyani et al. (2025) show that the optimal solution to this problem is achieved by first constructing a prediction set $C(X)$ with valid coverage, and then acting according to the max-min decision rule: $a(X) = \arg\max_{a \in \mathcal{A}} \min_{y \in C(X)} u(a, y)$. This result establishes that conformal prediction sets are a sufficient representation of uncertainty for risk-averse agents seeking a marginal notion of safety (1).

While marginal safety offers a useful baseline, *it fails to guarantee that any particular decision is safe*—it only ensures that, on average across the population, the agent avoids low-utility outcomes. In safety-critical settings, this average guarantee can be dangerously misleading: a high-risk action might appear acceptable because its risks are averaged out by safer decisions elsewhere. From the perspective of a downstream decision maker, what matters is not marginal safety, but whether each potential action carries a reliable safety guarantee. For instance, in medical treatment planning, a decision policy may appear safe on average but fail to provide safety when high-risk actions like surgery are chosen, making action-conditional guarantees essential. This motivates the need for *action-conditional safety*: a stronger requirement that ensures the decided action will lead to high utility with high probability, conditioned on that action being taken. The action-conditional guarantees are practically meaningful and essential for safe deployment in domains such as healthcare, where decisions must be justified not globally, but per-action.

Formally, for an action policy $a : \mathcal{X} \to \mathcal{A}$, a utility function $u : \mathcal{A} \times \mathcal{Y} \to \mathbb{R}$, and a utility certificate $\nu : \mathcal{X} \to \mathbb{R}$, we introduce the action-conditional safety constraint:

$$\mathbb{P}(u(a(X), Y) \geq \nu(X) \mid a(X) = a) \geq 1 - \alpha, \quad \text{for all } a \in \mathcal{A}. \tag{2}$$

This constraint ensures that each action the agent may take leads, with high probability, to utility exceeding the certificate $\nu(X)$ (larger certificates are desirable). It also draws a conceptual parallel to the *multi-calibration* literature Foster & Vohra (1997); Zhao et al. (2021); Noarov et al. (2023)—particularly *decision calibration*—which calibrates forecasts conditioned on the actions of downstream agents. However, such frameworks are tailored to *risk-neutral* decision makers who aim to maximize expected utility, for whom calibrated probability estimates are the correct notion of uncertainty quantification Foster & Vohra (1997); Zhao et al. (2021); Noarov et al. (2023). In contrast, the *risk-averse* agents we seek to avoid low-utility outcomes by optimizing quantiles of their utility distribution, and thus require *high-probability guarantees*. Our work can be viewed as an analogous foundation to decision calibration, adapted to the risk-averse setting.

In this work, we study such risk-averse agents who aim to maximize their average utility certificate $\mathbb{E}[\nu(X)]$, subject to satisfying the action-conditional safety guarantee in (2). This corresponds to agents optimizing their *action-conditional value-at-risk*—a formulation that captures the need for reliable utility under each possible decision. We will show that this objective admits a sharp characterization: the optimal decision policy can be implemented via conformal prediction sets that satisfy an *action-conditional validity* property:

$$\mathbb{P}\left(Y \in C(X) \mid \arg\max_{a \in \mathcal{A}} \min_{y \in C(x)} u(a, y) = a\right) \geq 1 - \alpha, \quad \forall a \in \mathcal{A}. \tag{3}$$

That is, prediction sets with the right conditional coverage structure serve as surrogates for action-conditional risk-averse decision making. Building on this line of thinking, we now detail our contributions to both sides of the story: decision making and conformal prediction.

**On the decision making side.** We prove that prediction sets with the right conditional coverage structure serve as surrogates for action-conditional risk-averse decision making. We establish a one-sided correspondence: optimizing action-conditional prediction sets yields feasible risk-averse policies. This strengthens the results of Kiyani et al. (2025) and further solidifies the role of conformal prediction as a principled uncertainty quantification tool—even when conditional safety guarantees are required in downstream decisions. Moreover, we explicitly derive the *optimal prediction sets* that achieve the best population-level performance under action-conditional constraints.

**On the conformal prediction side.** To translate this population-level characterization into a practical algorithm, we develop a novel *finite-sample debiasing method* that ensures distribution-free action-conditional coverage guarantees—marking the first such result in the literature. Our calibration procedure builds on and extends the methodology of Gibbs et al. (2025), by defining the novel notion of action specific non-conformity score. We also establish both *distribution-free lower and upper bounds* for the action-conditional coverage of our method.

**Experimental Validation.** We empirically evaluate AC-RAC on real-world tasks such as medical diagnosis. Across all settings, AC-RAC achieves significantly lower miscoverage for each action class compared to baseline methods, while maintaining competitive utility. These results highlight the practical relevance of action-conditional safety in decision-sensitive environments.

## 1.1 RELATED WORK

The formal framework of conformal prediction (CP) was introduced by Vovk et al. (1999); Saunders et al. (1999); Vovk et al. (2005). CP has since become a widely adopted method for constructing prediction sets with finite-sample marginal validity guarantees (for instance look at Shafer & Vovk (2008); Angelopoulos et al. (2023)). A large body of work has sought to strengthen the marginal guarantees of CP. These include group-conditional coverage Barber et al. (2023); Cauchois et al. (2021); Gibbs et al. (2025); Jung et al. (2022); Vovk et al. (2005), localized guarantees Guan (2023); Hore & Barber (2023), label-conditional coverage Ding et al. (2023); Vovk et al. (2005), selection-conditional coverage Bao et al. (2024); Jin & Ren (2025); Gazin et al. (2025); Jin & Candès (2023), and approaches for CP under treatment-based selection in counterfactual inference Lei & Candès (2021); Yin et al. (2024); Jin et al. (2023).

Our work introduces a new notion: action-conditional coverage, which is fundamentally distinct. While action-conditional guarantees share similarities with group-conditional CP—specifically, in that they target coverage over groups shaped by the action policy—the key difference is that these groups are not predefined. They emerge dynamically from the decision policy, which itself depends on the prediction sets. This interdependence creates a feedback loop that breaks the usual predefine-then-calibrate pipeline and necessitates new algorithmic tools to ensure valid coverage across all actions. Similarly, selection-conditional methods address inference after a selection event in test time (e.g., coverage conditioned on the test points with small set size). However, our setting is richer: we require simultaneous coverage across all actions, not just for selected instances. Even in the binary action case (e.g., select vs. not-select), selection-conditional methods typically provide guarantees only for selected points, whereas we demand coverage for both selected and non-selected cases. Also beyond our contribution to conditional methods for debiasing sets, we also show that action-conditional prediction sets act as surrogates for risk-averse agents optimizing action-conditional value at risk, linking conditional CP with finer grained safety guarantees for decision making. For further discussion on related works, particularly on the risk averse decision making side of our contributions please see Appendix A.

## 2 ACTION-CONDITIONAL CONFORMAL RISK AVERSE DECISION MAKING

In this section, we will formulate the problem of risk-averse decision-making with action-conditional guarantees. We are given a space of features $\mathcal{X}$, a set of labels $\mathcal{Y}$, and a set of possible actions $\mathcal{A}$. We assume that $(x, y) \in \mathcal{X} \times \mathcal{Y}$ is drawn from an unknown joint distribution $\mathcal{D}$. After observing a feature $x$, the decision maker selects an action $a$. Note that the decision maker does not observe the true label $y$. However, the utility of the decision maker will depend on both the chosen action $a$ and the label $y$, and is captured by a given utility function $u : \mathcal{A} \times \mathcal{Y} \to \mathbb{R}$.

The core principle behind risk-averse optimization is to prioritize reliability over average performance. That is, rather than selecting actions that maximize expected utility, the agent prefers actions that avoid catastrophic low-utility outcomes with high probability. This is particularly important in safety-critical settings such as healthcare or finance, where occasional failures can be unacceptable. In the classical risk-averse setting, the agent aims to guarantee that the utility $u(a, Y)$ is high with probability at least $1 - \alpha$. For a pre-specified risk tolerance level $\alpha \in (0, 1)$, the $\alpha$-level quantile of the utility for each action $a \in \mathcal{A}$ is defined as

$$\nu_\alpha(a; x) = \text{quantile}_\alpha\left(u(a, Y) \mid X = x\right).$$

A risk-averse decision maker then selects the action that maximizes this guaranteed utility level—that is, the action whose worst-case performance, up to level $1 - \alpha$, is as high as possible. This leads to conservative decisions that avoid low-utility outcomes with high confidence.

This formulation requires full knowledge of the conditional distribution $p(y \mid x)$, which is typically inaccessible. To address this, Kiyani et al. (2025) introduce a marginal relaxation called RA-DPO (Risk-Averse Decision Policy Optimization), where the quantile constraint is enforced over the entire data distribution:

$$\max_{a(\cdot), \nu(\cdot)} \quad \mathbb{E}_X[\nu(X)],$$
$$\text{s.t.} \quad \mathbb{P}(u(a(X), Y) \geq \nu(X)) \geq 1 - \alpha, \tag{4}$$

where $a : \mathcal{X} \to \mathcal{A}$ is the policy and $\nu : \mathcal{X} \to \mathbb{R}$ is the associated utility certificate. This marginal formulation provides a statistically feasible proxy for pointwise quantile guarantees.

In this paper, we propose a stronger notion of risk-aversion: one that is *action-conditional*. That is, the utility guarantee must hold within each subpopulation of the data defined by the action taken. This is essential in applications where actions have distinct operational meanings (e.g., treatment decisions), and fairness or reliability must be certified at the level of each action.

Formally, we impose the requirement that for each action $a \in \mathcal{A}$, the agent achieves a high-utility outcome with probability at least $1 - \alpha$ conditional on selecting that action. This leads to the following optimization problem, termed as *Action-Conditional Decision Policy Optimization* (AC-DPO):

$$
\begin{aligned}
\max_{a(\cdot), \nu(\cdot)} \quad & \mathbb{E}_X[\nu(X)], \\
\text{s.t.} \quad & \mathbb{P}(u(a(X), Y) \geq \nu(X) \mid a(X) = a) \geq 1 - \alpha, \quad \forall a \in \mathcal{A}.
\end{aligned}
\tag{5}
$$

Here, recall that $u$ denotes the utility function, $a : \mathcal{X} \to \mathcal{A}$ is the decision policy, $\nu : \mathcal{X} \to \mathbb{R}$ is the quantile function, and $\alpha$ is the risk tolerance level. This formulation strengthens the guarantee in (4) by ensuring calibrated performance across all decision branches, not only in aggregate. We refer to (5) as the action-conditional counterpart to RA-DPO.

## 2.1 Constructing Action-Conditional Policies from Prediction Sets

Prediction sets have emerged as a central tool in uncertainty quantification due to their ability to provide distribution-free, model-agnostic coverage guarantees. Recent work by Kiyani et al. (2025) has established a fundamental connection between prediction sets and risk-averse decision making. In particular, it is shown that that any optimal decision policy under marginal safety constraints can be equivalently realized through a max-min decision rule over appropriately constructed prediction sets. Formally, it is shown that RA-DPO (4) is equivalent to the solution of a prediction-set-based optimization problem called Risk-Averse Conformal Prediction Optimization (RA-CPO),

$$
\begin{aligned}
\max_{C(\cdot)} \quad & \mathbb{E}_X \left[ \max_{a \in \mathcal{A}} \min_{y \in C(X)} u(a, y) \right] \\
\text{s.t.} \quad & \mathbb{P}(Y \in C(X)) \geq 1 - \alpha.
\end{aligned}
$$

Using an optimal solution $C(x)$ of this problem, we can find the optimal policy for (4) as

$$
a_{\mathrm{RA}}^C(x) = \arg\max_{a \in \mathcal{A}} \min_{y \in C(x)} u(a, y), \qquad \nu_{\mathrm{RA}}^C(x) = \max_{a \in \mathcal{A}} \min_{y \in C(x)} u(a, y).
\tag{6}
$$

This equivalence shows that prediction sets communicate uncertainty effectively and serve as a complete representation for optimizing risk-averse utility under marginal coverage constraints.

We now generalize this result to the *action-conditional* setting, where safety must hold *conditionally* on each action being taken. Specifically, we consider the *Action-Conditional Risk-Averse Conformal Prediction Optimization* (AC-CPO):

$$
\begin{aligned}
\max_{C(\cdot)} \quad & \mathbb{E}_X \left[ \max_{a \in \mathcal{A}} \min_{y \in C(X)} u(a, y) \right] \\
\text{s.t.} \quad & \mathbb{P}\left( Y \in C(X) \mid a_{\mathrm{RA}}^C(X) = a \right) \geq 1 - \alpha, \quad \forall a \in \mathcal{A}.
\end{aligned}
\tag{7}
$$

We next show that this action-conditional generalization leads to a feasible policy optimization formulation (defined in (5)). That is, prediction sets constitute a relaxed representation for risk-averse decision making when conditioning on individual actions.

**Theorem 1 (From AC-CPO to AC-DPO)** *From any optimal solution $C^*$ to (7), we can construct a feasible solution $(a_{\mathrm{RA}}^{C^*}(x), \nu_{\mathrm{RA}}^{C^*}(x))$ for (5) such that it follows the action-conditional constraints and*

$$
\mathbb{E}_X \left[ \nu_{\mathrm{RA}}^{C^*}(X) \right] = \mathbb{E}_X \left[ \max_{a \in \mathcal{A}} \min_{y \in C^*(X)} u(a, y) \right].
$$

We refer the reader to Appendix C.1 for the proof. In contrast to the two-way equivalence shown by Kiyani et al. (2025), our action-conditional setting yields only a one-directional guarantee: an optimal AC-CPO solution induces a feasible AC-DPO policy with equal value, whereas the converse

need not hold. This asymmetry arises because, under action-conditional constraints, the conditioning event in AC-CPO $\{a_{\mathrm{RA}}^C(X) = a\}$ is endogenous to the prediction sets but the event in AC-DPO $\{a(X) = a\}$ is free. Despite this relaxation, our experiments (Section 5) show that the resulting policies preserve high utility while delivering valid action-conditional safety. Accordingly, we use AC-CPO as a principled surrogate that preserves conditional guarantees and provides a constructive route to implementable policies.

# 3 REPARAMETERIZED CONSTRUCTION OF PREDICTION SETS

In this section, we study the population-level conformal prediction problem with action-conditional guarantees, as defined in (7). Our goal is to characterize the prediction set function $C(x)$ that maximizes risk-averse utility while satisfying per-action coverage constraints. To this end, we introduce a reparameterization and derive a dual formulation using tools from convex duality. See Theorem 3 for the final characterization. The principles developed here will serve as the foundation for designing a finite-sample method in the next section. All technical proofs are deferred to Appendix C.2. We begin by defining two quantities that encapsulate the risk-averse utility structure:

$$\theta(x,t) = \max_{a \in \mathcal{A}} \text{quantile}_{1-t}\left[u(a,Y) \mid X = x\right], \quad a(x,t) = \arg\max_{a \in \mathcal{A}} \text{quantile}_{1-t}\left[u(a,Y) \mid X = x\right]. \tag{8}$$

Here, $\theta(x,t)$ denotes the maximum achievable risk-averse utility at coverage level $t \in (0,1)$, and $a(x,t)$ denotes the action that attains this utility. Intuitively, we aim to assign a feature-dependent coverage threshold $t(x) \in [0,1]$, such that the resulting prediction set $C(x)$ satisfies action-conditional coverage and induces high utility. To this end, we reparameterize the prediction set problem (7) in terms of the pointwise coverage function $t(x) = \mathbb{P}(Y \in C(X) \mid X = x)$. We show that (7) can be relaxed to the following optimization problem over coverage functions:

$$\begin{aligned} \max_{t:\mathcal{X}\to[0,1]} \quad & \mathbb{E}_X\left[\theta(X, t(X))\right], \\ \text{s.t.} \quad & \mathbb{E}\left[t(X) \mid a(X, t(X)) = a\right] \geq 1 - \alpha, \quad \forall a \in \mathcal{A}. \end{aligned} \tag{9}$$

The next proposition makes this relaxation precise and gives a closed-form expression for the prediction sets:

**Proposition 2 (Set Characterization)** *From any optimal solution $t^*(x)$ to (9), we obtain a feasible coverage function $C^*(x)$ for (7) such that*

$$\mathbb{E}_X\left[\max_{a \in \mathcal{A}} \min_{y \in C^*(X)} u(a,y)\right] = \mathbb{E}_X\left[\theta(X, t^*(X))\right].$$

*The prediction set and decision policy are given by:*

$$C^*(x) = \{y \in \mathcal{Y} : u(a(x, t^*(x)), y) \geq \theta(x, t^*(x))\}, \qquad a^*(x) = a(x, t^*(x)). \tag{10}$$

*Moreover, this prediction set satisfies $t^*(x) = \mathbb{P}(Y \in C^*(X) \mid X = x)$.*

This reparameterization serves as a conceptual bridge between the prediction set formulation and the risk-averse decision policy. Once the optimal function $t^*(x)$ is obtained, we can construct both the feasible prediction sets and the associated actions via (10). We now turn to solving (9). Unlike the marginal setting in Kiyani et al. (2025), where the coverage constraint is convex and decoupled from the decision rule, the action-conditional formulation introduces new complexity: the conditioning event $a(X, t(X)) = a$ depends nonlinearly on the decision variable $t$. This dependence breaks convexity and prevents direct application of previous techniques. We begin by rewriting the action-conditional constraint in (9) using the identity

$$\mathbb{E}_X[t(X)\mathbb{1}_a(X, t(X))] \geq (1 - \alpha)\mathbb{P}(a(X, t(X)) = a),$$

where $\mathbb{1}_a(x,t) = \mathbb{1}[a(x,t) = a]$ denotes the indicator that action $a$ is selected under threshold $t$. This reformulation casts both sides of the constraint as expectations, enabling a more linear treatment of the dependence on $t$ and the decision rule $a(x,t)$.

A key challenge, however, is that the indicator function $\mathbb{1}_a(x,t)$ introduces a discontinuous dependence on $t$, leading to nonconvexity of the problem. We handle this difficulty by developing new techniques detailed in Appendix C.2.2, drawing on tools from duality theory. To proceed, we define

the threshold function $t^*(x, \boldsymbol{\lambda})$ using a family of parameters $\boldsymbol{\lambda} = (\lambda_a)_{a \in \mathcal{A}} \in \mathbb{R}_{\geq 0}^{|\mathcal{A}|}$, one for each action

$$t^*(x, \boldsymbol{\lambda}) = \arg\max_{t \in [0,1]} \left\{ \theta(x, t) + \sum_{a \in \mathcal{A}} \lambda_a (t - (1 - \alpha)) \mathbb{1}_a(x, t) \right\}. \tag{11}$$

This expression captures the tradeoff between utility and action-conditional coverage through a tunable vector of $|\mathcal{A}|$-dimensional parameters $\boldsymbol{\lambda}$. The next theorem proves that this formulation can fully describe the optimal solution to the problem (9).

**Theorem 3 (Strong Duality)** *Assume the marginal distribution $\mathcal{P}_X$ is continuous. Then strong duality holds for the reparameterized problem (9). In particular, there exists $\boldsymbol{\lambda}^* \in \mathbb{R}_{\geq 0}^{|\mathcal{A}|}$ such that*

$$t_{\mathrm{opt}}(x) = t^*(x, \boldsymbol{\lambda}^*)$$

*is the optimal solution to (9). Moreover, the dual minimizer is given by:*

$$\boldsymbol{\lambda}^* = \arg\min_{\boldsymbol{\lambda} \in \mathbb{R}_{\geq 0}^{|\mathcal{A}|}} \Psi(\boldsymbol{\lambda}), \quad where \quad \Psi(\boldsymbol{\lambda}) = \mathbb{E}_X \left[ \max_{t \in [0,1]} \left\{ \theta(X, t) + \sum_{a \in \mathcal{A}} \lambda_a (t - (1 - \alpha)) \mathbb{1}_a(X, t) \right\} \right].$$

This result provides a tractable characterization of problem (9) with action-dependent constraints.

## 4 ACTION-CONDITIONAL RISK AVERSE CALIBRATION

In this section, we consider the risk-averse optimization problem in the finite-sample setting. Suppose that we have access to a set of calibration samples $\{(x_i, y_i)\}_{i \in [n]}$ and a predictive model $f : \mathcal{X} \to \Delta(\mathcal{Y})$, which assigns each $x \in \mathcal{X}$ to a probability vector in $\Delta(\mathcal{Y})$. We denote the output of $x$ as $f_x$, which is an approximation of the distribution of the label $y$ given the input $x$. In practice, $f$ is the (softmax) output of a pre-trained model that predicts label $y$ from $x$. In this section, we aim to develop a finite sample algorithm that connects predictions to actions that can exploit any black-box pre-trained predictive model. We present all the technical proofs in Appendix C.3.

Given the model $f$, we estimate functions $\theta$ and $a$ defined in (8), via replacing the true conditional probabilities with their estimated counterparts obtained by $f$. In more detail, we obtain $\hat{\theta}$ and $\hat{a}$ via

$$\hat{\theta}(x, t) = \max_{a \in \mathcal{A}} \mathrm{quantile}_{1-t} \big[ u(a, Y) \mid Y \sim f_x \big], \quad \hat{a}(x, t) = \arg\max_{a \in \mathcal{A}} \mathrm{quantile}_{1-t} \big[ u(a, Y) \mid Y \sim f_x \big]. \tag{12}$$

For any $\boldsymbol{\lambda} \in \mathbb{R}_{\geq 0}^{|\mathcal{A}|}$, define the optimal assignment function

$$\hat{t}(x, \boldsymbol{\lambda}) = \arg\max_t \left( \hat{\theta}(x, t) + \sum_a \lambda_a (t - (1 - \alpha)) \mathbb{1}_{\hat{a}(x,t)=a} \right). \tag{13}$$

and the conformal prediction set $\hat{C}(x, \boldsymbol{\lambda}) = \{ y \in \mathcal{Y} : u(\hat{a}(x, \hat{t}(x, \boldsymbol{\lambda})), y) \geq \hat{\theta}(x, \hat{t}(x, \boldsymbol{\lambda})) \}$.

We note that all the functions can be easily computed because they are parametrized by $\boldsymbol{\lambda}$. From Theorem 3 we know that there exists a $\boldsymbol{\lambda}^*$ such that the optimal prediction sets is derived using the function $\hat{t}(x, \boldsymbol{\lambda}^*)$ and $\hat{C}(x, \boldsymbol{\lambda}^*)$. Thus, the next question is now to select a feasible $\boldsymbol{\lambda}^*$ that satisfies the action-conditional coverage constraints. To do this, we first reparametrize the event $\{Y \in \hat{C}(x, \boldsymbol{\lambda})\}$ by a closed-form of $\boldsymbol{\lambda}$. In more detail, we note that $\hat{C}$ is determined by the assignment function $\hat{t}$, which can be decoupled by its one-dimensional counterpart. For any $a \in \mathcal{A}$, define $\hat{t}(x, \lambda_a)$ as

$$\hat{t}(x, \lambda_a) = \arg\max_{t \in [0,1]} \big( \hat{\theta}(x, t) + \lambda_a (t - (1 - \alpha)) \big),$$

then $\hat{t}(x, \lambda_a)$ is only parametrized by the scalar $\lambda_a$. We further define the action specific nonconformity score $\lambda_a^*$ for any $(x, y)$ as follows

$$\lambda_a^*(x, y) = \inf \left\{ \lambda_a \geq 0 : y \in \mathrm{QuantileSet}_{1-\hat{t}(x, \lambda_a)} [u(a, Y) \mid Y \sim f_x] \right\}.$$

where the quantile set is defined as

$$\mathrm{QuantileSet}_{1-t}[u(a, Y) \mid Y \sim f_x] = \left\{ y \in \mathcal{Y} : \mathbb{P}_{Y \sim f_x}(u(a, Y) \leq u(a, y)) \leq 1 - t \right\}.$$

Intuitively, the larger the value of $\lambda_a^*(x, y)$, the more conservative it is to select action $a$ at input $x$ when facing outcome $y$. This quantifies how atypical $y$ is for $x$, if we want to play action $a$. The lemma below, under mild tie-breaking assumptions, describes an important property of these scores.

**Lemma 4** *Conditioning on the event* $\{\hat{a}(x, \hat{t}(x, \boldsymbol{\lambda})) = a\}$, *the following events are equivalent*

$$\{Y \in \hat{C}(x, \boldsymbol{\lambda})\} = \{\lambda_a \geq \lambda_a^*(x, Y)\}.$$

Lemma 4 implies membership in the set $\hat{C}(x, \boldsymbol{\lambda})$ reduces to the scalar threshold test $\{\lambda_a \geq \lambda_a^*(x, Y)\}$ conditioned on picking action $a$. In other words, for each $(x, y)$, the action specific nonconformity score $\lambda_a^*(x, y)$, is the value such that $y$ lies in the prediction set exactly when $\lambda_a \geq \lambda_a^*(x, y)$.

To calibrate the $|\mathcal{A}|$-dimensional vector $\boldsymbol{\lambda}$, we propose a novel formulation based on pinball loss minimization using action specific nonconformity scores. Specifically, we include an imputed test pair $(x_{\text{test}}, y)$ alongside the $n$ calibration samples $\{(x_i, y_i)\}_{i=1}^n$, resulting in $n + 1$ total points. For each sample $i = 1, \ldots, n$ and the test point, we define the loss:

$$F_y(\boldsymbol{\lambda}) = \frac{1}{n+1} \sum_{i=1}^n \ell_\alpha \left( \sum_a \lambda_a \mathbb{1}_{\{\hat{a}(x_i, \hat{t}(x_i, \boldsymbol{\lambda})) = a\}}, \sum_a \lambda_a^*(x_i, y_i) \mathbb{1}_{\{\hat{a}(x_i, \hat{t}(x_i, \boldsymbol{\lambda})) = a\}} \right)$$

$$+ \frac{1}{n+1} \ell_\alpha \left( \sum_a \lambda_a \mathbb{1}_{\{\hat{a}(x_{\text{test}}, \hat{t}(x_{\text{test}}, \boldsymbol{\lambda})) = a\}}, \sum_a \lambda_a^*(x_{\text{test}}, y) \mathbb{1}_{\{\hat{a}(x_{\text{test}}, \hat{t}(x_{\text{test}}, \boldsymbol{\lambda})) = a\}} \right),$$

where the pinball loss is defined as $\ell_\alpha(u, v) = (v - u)(\mathbb{1}_{u \leq v} - \alpha)$. Minimizing $F_y(\boldsymbol{\lambda})$ over $\boldsymbol{\lambda} \in \mathbb{R}_{\geq 0}^{|\mathcal{A}|}$ thus yields a set of parameters that encode action-specific thresholds, leading to finite-sample coverage guarantees. We can now present our main finite sample algorithm in 1.

---

**Algorithm 1** Action-Conditional Risk Averse Calibration (AC-RAC)

---

**Input:** Miscoverage level $\alpha$, calibration samples $\{(x_i, y_i)\}_{i=1}^n$, test covariate $x_{\text{test}}$
**for** *each* $y \in \mathcal{Y}$ **do**
$\quad \left\lfloor \quad \hat{\boldsymbol{\lambda}}_y = \arg\min_{\boldsymbol{\lambda} \in \mathbb{R}_{\geq 0}^{|\mathcal{A}|}} F_y(\boldsymbol{\lambda}). \right.$
**Output:** $C_{\text{final}}(x_{\text{test}}) = \{y \in \mathcal{Y} \mid y \in \hat{C}(x_{\text{test}}, \hat{\boldsymbol{\lambda}}_y)\}$

---

**Remark 5** *To solve the optimization problem of* $F_y(\boldsymbol{\lambda})$*, we can apply a gradient descent algorithm. To do so, one-can derive a sub-gradient* $F_y(\boldsymbol{\lambda})$*. We present the full algorithmic details in Algorithm 2, and provide a rigorous derivation of the sub-gradient step and convergence analysis in Appendix C.4.*

Next, we show that AC-RAC outputs prediction sets with distribution free guarantees at test time.

**Theorem 6 (Finite-Sample Action-Conditional Validity)** *Let* $(x_1, y_1), \ldots, (x_n, y_n), (x_{\text{test}}, y_{\text{test}})$ *be exchangeable samples. Then for every* $a \in \mathcal{A}$,

$$\mathbb{P}\big(y_{\text{test}} \in C_{\text{final}}(x_{\text{test}}) \mid a_{\text{RA}}^{C_{\text{final}}}(x_{\text{test}}) = a\big) \geq 1 - \alpha.$$

*If* $(x_i, y_i)$ *are i.i.d. and the score* $\lambda_a^*(X, Y)$ *has a continuous distribution for all* $a$, *then*

$$\mathbb{P}\big(y_{\text{test}} \in C_{\text{final}}(x_{\text{test}}) \mid a_{\text{RA}}^{C_{\text{final}}}(x_{\text{test}}) = a\big) \leq 1 - \alpha + \frac{|\mathcal{A}|}{(n+1) \cdot \mathbb{P}(a_{\text{RA}}^{C_{\text{final}}}(x_{\text{test}}) = a)} \quad \text{for each } a \in \mathcal{A}.$$

Theorem 6 guarantees that the proposed method achieves valid action-conditional coverage with finite samples and the coverage error is tightly controlled and decays at a rate of $1/n$. The i.i.d. and continuity condition is purely a tie-breaking device: it guarantees that, with probability 1, at most one calibration point per action lands exactly on the set boundary, which we need to bound the slack term in the upper–tail inequality. Such tie-breaking assumptions are standard in finite-sample conformal-prediction bounds and can always be enforced in practice by adding an arbitrarily small random jitter to the conformity scores, without affecting coverage or utility. We remark that our proof steps are inspired by Gibbs et al. (2025) and the results are analogous (see Theorem 3 in Gibbs et al. (2025)).

**Corollary 7** *Assume that* $(x_1, y_1), \ldots, (x_n, y_n), (x_{\text{test}}, y_{\text{test}})$ *are exchangeable. We then have*

$$\mathbb{P}\left( u\left( a_{\text{RA}}^{C_{\text{final}}}(x_{\text{test}}), y_{\text{test}} \right) \geq \nu_{\text{RA}}^{C_{\text{final}}}(x_{\text{test}}) \mid a_{\text{RA}}^{C_{\text{final}}}(x_{\text{test}}) = a \right) \geq 1 - \alpha, \quad \forall a \in \mathcal{A}.$$

Corollary 7 ensures that the max-min decision policy applied to the final prediction set $C_{\text{final}}$ yields a distribution-free, action-conditional safety guarantee. Specifically, the utility achieved by the selected action exceeds the certified risk-averse threshold with probability at least $1 - \alpha$, conditional on the action taken. Combined with Theorem 1, this result confirms that the output of our finite-sample algorithm satisfies the original action-conditional risk-averse objective in (7), thereby closing the loop between population-level theory and implementable practice.

## 5 NUMERICAL EXPERIMENTS

In this section, we empirically evaluate the performance of the proposed action-conditional algorithm (denoted by AC-RAC) on two experimental setups. We begin by describing the competing baselines, followed by the evaluation metrics. We compare AC-RAC against two families of methods, each instantiated with the same pre-trained probabilistic model $f \colon (x, y) \mapsto f_x(y)$, where $f_x(y)$ is the assigned probability of the input-label pair $(x, y)$.

**Baseline 1: Conformal Prediction with Max-Min Decision Rule.** We benchmark our method against established conformal inference techniques by generating $(1 - \alpha)$-valid prediction sets $C(x)$ using split conformal prediction. Among these, we include the `RAC` method of Kiyani et al. (2025), which represents the state-of-the-art approach for risk-averse decision making under *marginal* safety guarantees. `RAC` employs a risk-sensitive scoring function to construct prediction sets independent of the action space, followed by a max-min decision rule that ensures robust performance against worst-case outcomes. In addition to `RAC`, we include two action-independent conformal baselines. `score-1` (Sadinle et al., 2019): inverse probability score, defined as $1 - f_x(y)$; `score-2` (Romano et al., 2020): cumulative tail mass, $\sum_{y' : f_x(y') > f_x(y)} f_x(y')$. All methods yield a prediction set $C(x)$, which is subsequently paired with a max-min utility decision rule: $a_{\mathrm{RA}}^C(x) = \arg\max_{a \in \mathcal{A}} \min_{y \in C(x)} u(a, y)$, selecting the safest action that maximizes worst-case utility across the set.

**Baseline 2: Calibrated Best-Response.** We also evaluate against a calibration-based decision-making baseline. This approach applies the decision calibration procedure of Noarov et al. (2023) to adjust the predictive model on a held-out calibration set, ensuring improved reliability through swap regret bounds introduced in Zhao et al. (2021). Once the calibrated forecast $f_x(y)$ is obtained, actions are selected to maximize expected utility: $a_{\mathrm{BR}}(x) = \arg\max_{a \in \mathcal{A}} \mathbb{E}_{y \sim f_x}[u(a, y)]$. This standard risk-neutral method performs well on average utility but lacks coverage guarantees and is vulnerable to high-risk errors under underestimated uncertainty.

**Evaluation Metrics.** For methods in Baseline 1, we evaluate performance using the following two metrics. **Action-specific miscoverage:** the empirical miscoverage rate conditioned on each selected action, assessing conditional reliability; **Worst-case utility:** the average realized utility under the max-min rule, $\mathbb{E}[\max_a \min_{y \in C(x)} u(a, y)]$, capturing conservative performance. In addition, to compare all methods (include Calibrated Best-Response) we report **Critical error rate**, which is the fraction of samples where the selected action leads to a highly adverse utility outcome, indicating the frequency of catastrophic decisions. Additional evaluations, including average utility and sweeps over multiple $\alpha$ values for action-specific miscoverage, are presented in Appendix D.

### 5.1 MEDICAL DIAGNOSIS

We evaluate AC-RAC in the context of risk-sensitive medical diagnosis and treatment planning. Our dataset is the COVID-19 Radiography Database (Chowdhury et al., 2020; Rahman et al., 2021), comprising chest X-ray images labeled as *Normal*, *Pneumonia*, *COVID-19*, or *Lung Opacity*. Images are partitioned at random into training (70%), calibration (10%), and test (20%) subsets.

For feature extraction, we use Inception-v3 (Szegedy et al., 2015; 2016) pretrained on ImageNet, fine-tuned from the second inception block. Clinical trade-offs are encoded via a utility matrix (Table 1) mapping each diagnosis to a set of actions. While we report results using this matrix, our framework supports alternative specifications (see Appendix B of Kiyani et al. (2025)). All baselines are calibrated to ensure consistent mapping of model outputs to the four actions.

Figure 1 (top row) presents the results of the medical diagnosis task at a nominal miscoverage level of $\alpha = 0.05$. The left panel shows that AC-RAC is the only method achieving valid conditional coverage across all actions, while `RAC`, `score-1`, and `score-2` systematically over- or under-cover. The middle panel reports the average realized max-min utility across $\alpha \in \{0.01, 0.02, 0.03, 0.05, 0.1\}$, revealing that AC-RAC achieves higher worst-case utility than `score-1` and `score-2`, and slightly lower utility than `RAC`. This gap is expected, as `RAC` is optimized for marginal coverage, while AC-RAC imposes stricter action-conditional guarantees, which introduces a modest trade-off in utility. The right panel shows the rate of critical errors, where a high-risk action is selected for a vulnerable patient group. Here, AC-RAC reduces harmful decisions by a significant margin, highlighting its advantage in safety-critical settings like clinical decision-making. We remark that the left plot only contains action 0,1,3 instead of action 2 because `score-1`,`score-2` and `RAC` never select action 2.

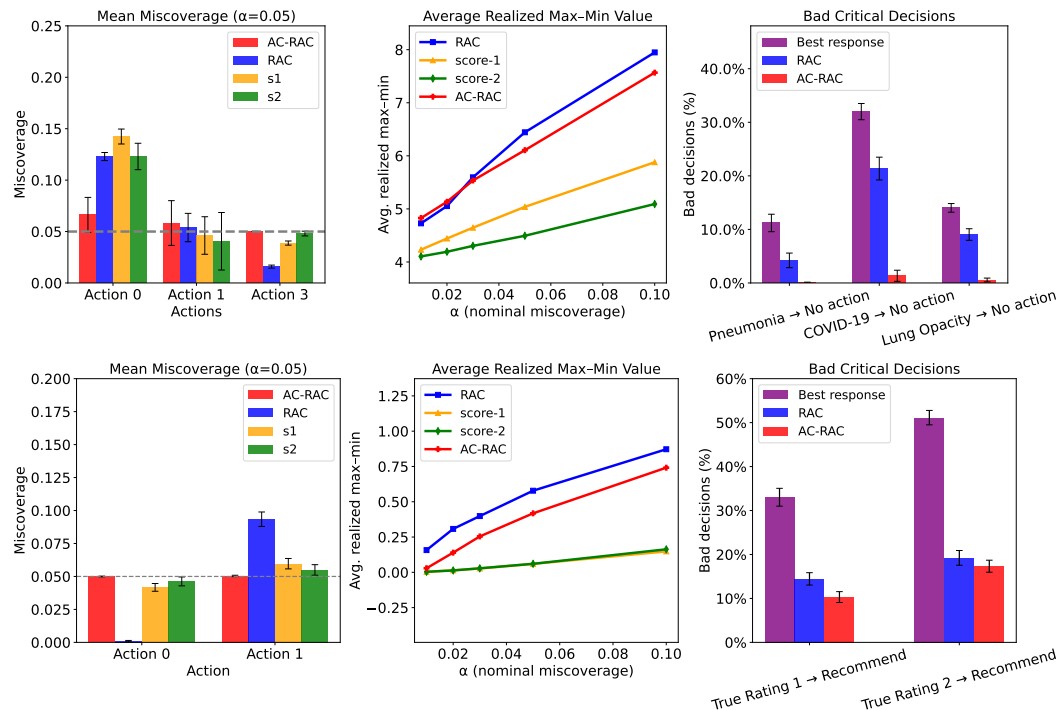

Figure 1: Comparison of methods under action-conditional constraints at nominal level $\alpha = 0.05$. Each row shows (left) mean miscoverage across actions, (center) average realized max-min utility, and (right) critical decision error rate. The first row is the result of medical diagnosis and second row is recommender system. All the error bars are averaged over 40 random seeds.

## 5.2 RECOMMENDER SYSTEMS

We next evaluate AC-RAC in the context of recommendation systems, where the goal is to decide about whether to suggest a movie to a user. Each feature corresponds to a user–item pair $x = (u, m)$, where $u$ and $m$ are feature vectors representing the user and the movie, respectively. The associated label $y \in \{1, 2, 3, 4, 5\}$ denotes the rating assigned by the user. The action space is binary: $\mathcal{A} = \{\text{No-Rec, Rec}\}$, where Rec indicates recommending the movie. We partition the dataset into three splits: 80% for training the predictive model, 10% for calibration, and 10% for evaluation.

At decision time, utility of each action depends on the true rating (specified in Table 2). Recommending a movie with rating $y$ yields utility $u(\text{Rec}, y) = y - 3$, encouraging recommendations only when the rating is above average. The no-recommendation action has utility $u(\text{No-Rec}, y) = 0$, reflecting a conservative fallback. This setup captures the trade-off between opportunity and risk: aggressive recommendations can yield high returns but carry greater downside under uncertainty.

Figure 1 (bottom row) summarizes the performance of each method on a recommendation task under $\alpha = 0.05$. The left panel confirms that AC-RAC uniquely achieves calibrated action-conditional coverage, whereas the other methods suffer from miscalibration. This misalignment between coverage and decision type leads to reliability gaps. The middle panel shows that AC-RAC achieves worst-case utility comparable to RAC, while offering significantly better action-conditional coverage. The right panel reveals that AC-RAC substantially reduces the rate of critical recommendation errors.

## 6 CONCLUSION AND DISCUSSION

We introduced CP sets with action-conditional guarantees, and then showed these sets provide a feasible policy for risk averse decision makers seeking action conditional safety guarantees. Building upon that we construct finite-sample CP sets with provable action-conditional validity. While this work provides new avenues for principled uncertainty quantification in safety-critical domains, it will be interesting to explore other notions of safety beyond quantiles (such as CVaR).

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
