# OpenReview forum: "Conformal Risk-Averse Decision Making with Action Conditional Guarantees"
_ICLR.cc/2026/Conference — ICLR 2026 Conference Withdrawn Submission_

### Official Review · Reviewer_PgvK · 2025-10-27

**Soundness:** 3
**Presentation:** 2
**Contribution:** 2
**Rating:** 2
**Confidence:** 4

**Summary:**

This paper extends results presented by Kiyani et al. (2025), and introduces a notion of action conditional safety guarantees, characterized by a pre-specified utility function: the utility of a decision must exceed a utility certificate with high probability for each action. The authors propose a policy which can satisfy such guarantees using prediction sets with action-conditional coverage properties. In order to achieve action-conditional coverage, they propose a novel non-conformity score and an optimization-based method to identify a threshold for creating these prediction sets. They validate their results on Medical diagnosis and Movie recommendation tasks.

**Strengths:**

The paper makes a convincing case for action-conditional safety guarantees, and thus, it is easy for the reader to believe that the authors are tackling an important problem. The authors support their method with ample theoretical contributions. The experiments are sound and successfully strengthen the method.

**Weaknesses:**

The validity of the theoretical contributions is weakened by the lack of proofs in the main paper or the supplement. The structure of the paper and its results were largely modeled on Kiyani et al. (2024); however, there wasn’t enough background on that paper to contextualize these results. I found it very difficult to understand the purpose of some results without referencing their analogous results in Kiyani et al. (2024).  I believe the paper would benefit from a section dedicated to a comprehensive review of Kiyani et al. (2024). It’s also not clear how $\lambda^*(x, y)$ intuitively behaves like a non-conformity score. How does a larger $\lambda$ imply the atypicality of $y$ being paired with $x$ for a given action? On the experiments side, the authors could’ve included a calibration plot to show the mean empirical miscoverage for various $\alpha$ levels (this was also present in the Kiyani et al. (2024)).

**Questions:**

* What are Action 0, Action 1, and Action 3 in Figure 1?
* What is the purpose of Lemma 4? Does this equivalence provide ease in constructing prediction sets? Does it motivate the use of a pinball loss?
* In the case of conformal prediction, we see that the quality of $f$ affects the size of the conformal prediction sets. How does the quality of $f$ affect the guarantee? Does it reduce $\nu(x)$? If so, how?
* What is the utility function for the medical diagnosis task?

---

### Official Review · Reviewer_BFp8 · 2025-10-31

**Soundness:** 4
**Presentation:** 4
**Contribution:** 2
**Rating:** 4
**Confidence:** 3

**Summary:**

This paper proposes an extension on the conformal risk-averse decision making. The action-conditional property is useful in decision-making scenarios. The paper shows that the optimal risk-averse decisions are equivalent to the minimax decisions when the ground truth is chosen from the conformal set. The technical parts of the paper include an algorithm to get risk-averse calibration. The authors validate the results by a numerical study, show the trade-off between between marginal and conditional reliability of conformal set.

**Strengths:**

- The motivation of the paper is clear. The distinction between population-level and per-action reliability is meaningful, especially in high-stake decision-making scenarios.
- The empirical results provide interesting results. The results demonstrate the trade-off between marginal and conditional reliability and provide intuitions for using the proposed method.

**Weaknesses:**

- Techniques closely follow Kiyani et al. 2025, making the technical novelty questionable.
- AC-RAC requires per-test optimization over $\lambda$ for each possible label and action, which may be computationally intensive. The paper lacks complexity analysis, runtime benchmarks, or ablation studies.

**Questions:**

- What is the computational complexity of AC-RAC with respect to $|A|$ and $|\mathcal{Y}|$? If possible, can the authors explain more on how it scales to modern deep learning models?

---

### Official Review · Reviewer_BVCY · 2025-11-01

**Soundness:** 3
**Presentation:** 4
**Contribution:** 4
**Rating:** 2
**Confidence:** 4

**Summary:**

The paper proposes a novel notion of "action conditional guarantees" in the intersection of conformal prediction and decision theory.
This is somewhat analogous to existing notions of e.g. label conditional guarantees and group conditional guarantees, but directly targets the decision policy to be taken over the produced predictive sets.
When adopting a decision-theoretic view of conformal prediction this becomes a natural target, as the authors argue.
The authors develop this idea in the paper, starting from the decision theoretic view and, through significant manipulation, arriving at a readily implementable procedure with strong finite-sample properties (as usual for conformal methods) that provably satisfies action-conditional guarantees under a minimax policy.
This algorithm is shown to have superb empirical performance.

**Strengths:**

The paper is quite strong, and I enjoyed reading it.
The problem being tackled is interesting and of great relevance to both the community and practitioners, and the derivations proposed in the paper are reasonably nontrivial.

As far as I can tell, action conditional guarantees are outside of the reach of the usual techniques used for e.g. group conditional and label conditional guarantees, fundamentally requiring the through decision-theoretic view that the authors take.

**Weaknesses:**

I actually see very few weaknesses in the paper, and they are minor:

- The calibration procedure described in Algorithm 1 seems a bit more costly than many usual conformal calibration procedures. For one it is analogous to label conditional conformal prediction in that it requires an independent calibration for each target $y \in \mathcal{Y}$ (and thus requires $\mathcal{Y}$ to be finite), but this calibration now requires the use of a subgradient descent algorithm rather than a single quantile computation. Could the authors discuss the efficiency of this procedure? Perhaps an interesting point of comparison here be the work of [Gibbs et al., 2025].
- In the numerical experiments, it would be nice to also compare the proposed method to label conditional and group conditional split conformal prediction (especially label conditional). It might also be of interest to compare to some method of [Gibbs et al., 2025]. Perhaps naively, I expect that label conditional split conformal will always satisfy coverage, but will achieve a lower utility than the proposed method.

**HOWEVER:**
Unfortunately, it appears that the authors have (hopefully accidentally) not submitted the intended supplementary material. The PDF makes many references to appendices, e.g. lines 133/134, 214 and 230/231, but they are not present either in the PDF or as supplementary material on OpenReview.
This is troubling as the proofs to the theorems are supposedly present in Appendix C -- which the main text suggests is reasonably involved -- but which are not currently available for review.

While the theorems in the main text do seem reasonable and I can somewhat imagine most of their proofs, I find myself unable to recommend acceptance without reviewing them.
So while I would normally give a score of 'accept', I find myself obligated to give a score of 'reject'.
I strongly advise the authors to provide the proofs to be reviewed during the rebuttal phase, at which point I would be happy to update my score for acceptance. \
**Why not higher:** The lack of proofs in the currently submitted version hinders me from giving any score towards acceptance. This aside, the paper constitutes a meaningful contribution to the conformal prediction and uncertainty quantification literature, and I would be happy to recommend its acceptance. \
**Why not lower:** As I mentioned, were it not for the lack of proofs I would be happy to recommend acceptance. With this in mind the current score is already overly harsh, so it is undue to go any lower.

[Gibbs et al., 2025] Conformal prediction with conditional guarantees

**Questions:**

- I strongly suggest that the authors provide their proofs during the rebuttal period. If I am not mistaken they are able to directly update the PDF, which would perhaps be ideal. But if not, I suggest the proofs be submitted here on the OpenReview discussion, to be later included in the camera-ready version.
- I would appreciate it if the authors could also briefly comment on the two points I've raised in the 'weaknesses' section.

---

### Official Review · Reviewer_CRMx · 2025-11-01

**Soundness:** 2
**Presentation:** 2
**Contribution:** 2
**Rating:** 4
**Confidence:** 4

**Summary:**

The paper proposes a generalization of conformal prediction for risk-averse decision making, introducing action-conditional guarantees. It presents theoretical foundations, a dual formulation, and a finite-sample algorithm (AC-RAC) for constructing prediction sets that ensure safety conditioned on each action. Experiments on medical diagnosis and recommender systems demonstrate improved action-conditional reliability compared to marginal conformal baselines.

**Strengths:**

1. The shift from marginal to action-conditional guarantees is conceptually important for safety critical applications, where per action reliability is essential.
2. The paper provides formal definitions, duality results, and finite-sample guarantees, connecting conformal prediction to risk averse decision theory.
3. The AC-RAC algorithm is principled, distribution free, and leverages pinball loss minimization for calibration.

**Weaknesses:**

1. The paper does not adequately discuss or compare with the literature on conditional robust optimization (CRO), which also seeks decision rules with conditional guarantees and often provides more rigorous frameworks for uncertainty sets and risk measures (see e.g., Chenreddy & Delage, 2024; Patel et al., 2024b).
2. The action conditional framework only provides a one way correspondence (from prediction sets to feasible policies), unlike the two way equivalence in marginal CP. Is this a fundamental limitation? Can the gap be closed?
3. The AC-RAC algorithm involves gradient descent over action-specific nonconformity scores. How does this scale with large action spaces or high-dimensional data?
4. The experiments are compelling but limited to two domains. Broader empirical validation would strengthen the claims.

**Questions:**

1. How do you distinguish action-conditional conformal prediction from conditional robust optimization (CRO)? Are there scenarios where one framework is strictly preferable or more general than the other?
2. Could your action-conditional guarantees be extended to other risk measures, such as Conditional Value-at-Risk (CVaR) or mean-variance objectives? What challenges would arise in such generalizations.
3. In real-world deployments, how would you recommend practitioners interpret and use action conditional prediction sets?
4. Is the AC-RAC algorithm adaptable to online or streaming settings, where data arrives sequentially and decisions must be made in real time?
5. How sensitive is your approach to misspecification in the underlying predictive model? For example, if the model is poorly calibrated, does the action-conditional guarantee still hold?
6. Can action-conditional guarantees be leveraged to address fairness concerns, such as ensuring equitable safety across subpopulations or actions?
Are there intuitive ways to interpret the action-conditional prediction sets for stakeholders who may not be familiar with conformal prediction or risk-averse optimization?

---

### Note · Authors · 2025-12-02

**Comment:**

Due to a technical issue, the appendix for our submission did not upload successfully. We would therefore like to withdraw the paper from consideration.

**Withdrawal Confirmation:**

I have read and agree with the venue's withdrawal policy on behalf of myself and my co-authors.